# Long-Term Adverse Events Following Early Breast Cancer Treatment with a Focus on the *BRCA*-Mutated Population

**DOI:** 10.3390/cancers17152506

**Published:** 2025-07-30

**Authors:** Berta Obispo, Caroline Bailleux, Blanca Cantos, Pilar Zamora, Sachin R. Jhawar, Jajini Varghese, Lucia Cabal-Hierro, Paulo Luz, Luis Berrocal-Almanza, Xiaoqing Xu

**Affiliations:** 1Hospital Universitario Infanta Leonor, 28031 Madrid, Spain; berta.obispo@gmail.com; 2Centre Antoine Lacassagne, 06100 Nice, France; caroline.bailleux@nice.unicancer.fr; 3Hospital Universitario Puerta de Hierro, 28222 Madrid, Spain; blanca.cantos@salud.madrid.org; 4Hospital Universitario La Paz, 28046 Madrid, Spain; zamorapilar@gmail.com; 5Department of Radiation Oncology, The Ohio State University Comprehensive Cancer Center, Columbus, OH 43210, USA; sachin.jhawar@osumc.edu; 6The Stefanie Spielman Comprehensive Breast Center, Columbus, OH 43212, USA; 7The Breast Unit, UK Royal Free London NHS Trust, London NW3 2QG, UK; jajini.varghese1@nhs.net; 8Savana Research S.L., 28004 Madrid, Spain; lcabalhierro@savanamed.com (L.C.-H.); p_luz@msn.com (P.L.); 9Oncology Outcomes Research, AstraZeneca, Cambridge CB2 0AA, UK; luis.berrocal-almanza@astrazeneca.com; 10Oncology Outcomes Research, AstraZeneca, Gaithersburg, MD 20878, USA

**Keywords:** early breast cancer, (neo)adjuvant treatment, long-term toxicity, long-term adverse events, health-related quality of life, breast cancer survivorship, BRCA mutation, cancer treatment optimization

## Abstract

Assessing long-term adverse events (harmful or unexpected occurrences) in patients with early breast cancer is complex because of differences in characteristics across patient populations. This can be further complicated by variation in how ‘long-term’ adverse events are defined. Patients with inherited (germline) mutations in the *BRCA1* or *BRCA2* genes (gBRCAm) exhibit distinct types of adverse events, and specific assessment and management strategies may be required for these patients. New treatments present additional challenges, which may include new types of long-term adverse events. Doctors and carers should be aware of these potential events so that they can be effectively managed.

## 1. Introduction

Breast cancer (BC) is the most commonly diagnosed malignancy and cause of death in women worldwide, accounting for 2.3 million new cases (11.6% of all cancer cases) and 832,000 deaths in 2022. Biological factors, such as age, family history, and reproductive history, along with environmental factors like physical activity, alcohol consumption, weight, and hormonal supplements, have been recognized for their potential role in BC development [1]. Approximately 5–10% of BC cases have a family history, with 30% of these linked to mutations in genes predisposed to BC [2]. Approximately 10% of BC cases harbor a germline mutation in the breast cancer genes 1 (*BRCA1*) or 2 (*BRCA2*), (gBRCAm), which affects the homologous repair mechanism of DNA double-strand breaks. BRCA mutation carriers are diagnosed with BC at an early age (<40 years), and recent studies indicate that 55–65% of *BRCA1* and 45% of *BRCA2* mutation carriers will develop BC by age 70 [3,4]. However, most BC cases are sporadic, resulting from somatic mutations [5].

Standard treatments for early-stage BC (eBC), where the disease is confined to the breast or the axillary lymph nodes, may include surgery, chemotherapy, anti-HER2 therapy, and endocrine therapy (ET) [6,7]. Despite treatment, up to 20% of patients may experience recurrence within the first 10 years [8,9], particularly those with high-risk hormone receptor-positive (HR+) BC or with triple-negative BC (TNBC) [10,11,12,13].

Over 90% of BC cases are diagnosed early, and new therapies have significantly improved patients’ outcomes. Immunotherapy is effective for TNBC, while cyclin-dependent kinase 4 and 6 (CDK4/6) inhibitors have demonstrated benefits in HR+/HER2 BC. Furthermore, pembrolizumab, a programmed cell death protein 1 checkpoint inhibitor, has shown significant benefits in overall survival [14]. Doxorubicin plus cyclophosphamide enhances survival and response rates in patients with TNBC [15], while ribociclib and abemaciclib have improved disease-free survival rates [16,17]. Lastly, the poly(ADP)-ribose polymerase (PARP) inhibitor olaparib has improved survival in patients with gBRCAm/HER2-negative BC [18,19,20,21].

Updated guidelines for the diagnosis, treatment, and follow-up of eBC highlight that advancements in screening and therapeutic strategies have enhanced prognosis [22]. However, with 10-year disease-free survival rates ranging from 80 to 95% depending on subtype and stage, preserving patients’ health-related quality of life (HRQoL) post-treatment remains crucial [23,24]. While (neo)adjuvant therapies have been effective in reducing recurrence, they may cause long-term adverse events impacting physical, emotional, and psychosocial well-being [25,26].

Long-term adverse events are not well characterized, as there are no standardized definitions or timelines due to factors such as variability in regimens, patient physiology, and inconsistencies across registries. Consequently, there is no global consensus on the time frame defining an adverse event as early or long-term. In the literature, the time window for describing an adverse event as long-term varies widely, ranging from a few months to several years post-therapy. In addition, the definition of ‘long-term adverse event’ depends on factors such as treatment regimen, patient age and general health, and the specific outcome assessed.

Adverse events occurring from months to years post-therapy significantly impact HRQoL. Common complications include chronic pain (up to 72% of patients) [27], lymphoedema (27–40%) [28], peripheral neuropathy (23–80%), and/or gastrointestinal symptoms (42.8%) arising from chemotherapy treatment [29,30]. Additionally, conservative surgery combined with radiotherapy, while critical in adjuvant care, can lead to late cardiac effects and cosmetic challenges, despite advances aimed at reducing side effects [31].

Patients with eBC may also experience psychological alterations [32], fatigue [33], hormonal and endocrine disturbances [34,35], and sexual dysfunction and infertility [36]. These long-term adverse events also affect patients’ families, with financial strain and emotional stress often leading to depression and impacting recovery [37].

Long-term adverse events pose significant challenges for clinicians, patients, and families. This review examines these events, their risk factors, and management strategies, with a focus on patients with gBRCAm BC, aiming to equip clinicians and patients with the scientific knowledge and tools to optimize care and improve outcomes. Although long-term adverse events can be classified in various ways (e.g., by life impact, treatment modality, severity, or patient-reported outcomes), we have chosen to organize them according to symptoms grouped by affected body systems. This classification reflects how clinical care is typically structured, enabling more effective referral to relevant specialists, enhancing interdisciplinary collaboration, and supporting the development of targeted follow-up strategies. Our approach offers a practical lens for addressing the complex and multifaceted needs of survivors of BC. Despite its clinical relevance, this perspective is underrepresented in the literature, and this review aims to contribute to filling that gap.

## 2. Methodology

A systematic literature search was conducted in academic databases including PubMed, Embase, and the Cochrane Library, using the keywords ‘breast cancer’, ‘early breast cancer’, ‘long-term toxicity’, ‘adjuvant treatment’, ‘survivorship’, ‘quality of life’, and ‘BRCA mutation’. Articles published between January 1990 and May 2024, in English or Spanish, were included if they assessed, either retrospectively or prospectively, long-term adverse events in patients with eBC.

The search identified 335 articles, of which 243 were screened, and 130 were fully read (Figure 1). The findings of the review were synthesized and summarized in a narrative format.

## 3. Long-Term Adverse Events in Patients with eBC

Assessing long-term adverse events in eBC is challenging due to patient heterogeneity, varying treatments, comorbidities, and external factors such as lifestyle, genetic predisposition, or healthcare resources, which influence outcomes. The lack of standardized definitions and reporting methods further complicates comparisons across studies [38,39].

Establishing a consensus for a definition of ‘long-term’ adverse events is crucial, as it varies by symptom type (Table 1). For instance, ‘long-term’ hormonal therapy (HT) effects are monitored starting 1 month post-treatment [40], chronic pain after 3 months [41], and fatigue, memory, or psychological issues between 5 and 10 years [42,43]. Cardiac events may appear within a year or up to 8 years post-treatment [44,45,46].

To address these discrepancies, we reviewed studies on ‘long-term’ adverse events by symptom. The following sections outline key adverse events, their associated risk factors, and clinical management recommendations (see Table 2 for an overview).

### 3.1. Adverse Events Affecting the Chest Wall and Breast

Breast surgery and radiotherapy have the potential to cause various long-term adverse events in the chest wall and breast. The most commonly known are chronic pain (known as post-mastectomy pain syndrome or PMPS), lymphoedema, or skin and soft tissue fibrosis/necrosis related to radiotherapy [38].

#### 3.1.1. Post-Mastectomy Pain Syndrome

PMPS is a chronic pain condition caused by nerve fiber damage, resulting in a type of neuropathic pain. This ache is commonly felt in the anterior/lateral chest wall, axilla, and/or medial upper arm for a period exceeding 3 months post-surgery. Symptoms include neuropathic pain sensations such as burning, tingling, shooting, stinging, or stabbing pain, as well as hyperesthesia [62]. The reported incidence of nerve fiber damage and chronic pain following BC surgery ranges from 20% to 72% depending on the definition and duration of symptoms. Risk factors for persistent pain include postoperative pain, younger age, high body mass index (BMI), axillary radiation, and axillary lymph node dissection [27]. It is important to note that in recent years, surgical de-escalation strategies, such as an increased use of breast-conserving surgery and conservative mastectomies, have been widely adopted, which may influence the prevalence and severity of these long-term adverse events [75,76,77]. PMPS can have a significant negative impact on patients’ HRQoL, particularly their mental health, as indicated by retrospective studies [47,78]. Treatment options for PMPS include analgesics for neuropathic pain, surgical interventions such as neuroma excision or scar release, and complementary therapies like acupuncture or hypnosis [61].

#### 3.1.2. Lymphoedema

Lymphoedema is a chronic and often progressive condition characterized by the accumulation of lymphatic fluid in the interstitial tissues, and most commonly affects the arm, breast, chest wall, and shoulder on the treated side. It remains one of the most significant and burdensome long-term adverse effects of treatment for BC survivors, with a reported prevalence ranging from 27% to 40% depending on treatment type, time since surgery, and diagnostic criteria.

Current clinical practice emphasizes early risk stratification, patient education, and prompt initiation of conservative, non-surgical management. Axillary lymph node dissection (ALND) is the most consistently identified risk factor, followed by regional radiotherapy, higher BMI, postoperative infections, and trauma to the affected limb. Sentinel lymph node biopsy has largely replaced ALND in many early-stage cases, significantly reducing lymphoedema risk, though it does not eliminate it entirely.

Early detection of lymphoedema is essential and supported by routine surveillance methods and patient education on early symptoms. First-line treatment involves complex decongestive therapy, which includes manual lymphatic drainage, compression, exercise, and skin care. Supervised resistance training is now recognized as safe and effective [63,79].

Patient education is vital for prevention and long-term management, emphasizing self-care practices and the use of compression. Contemporary lymphoedema care is proactive, multidisciplinary, and patient-centered, aiming to reduce complications and improve HRQoL for BC survivors [48].

#### 3.1.3. Skin and Soft Tissue Affection

Radiotherapy can induce skin and soft tissue fibrosis and rarely necrosis, which could lead to poor cosmesis. In this regard, the most frequently observed complication following radiotherapy (up to 43% of patients) is radiation-induced fibrosis of the skin and subcutaneous tissue [50,80]. This type of fibrosis is typically observed in areas with overlapping treatment fields after breast-conserving surgery with postoperative radiotherapy or after mastectomy and radiotherapy, particularly in patients with breast implants [31,81].

### 3.2. Cardiotoxicity

Treatment for eBC, including radiotherapy, chemotherapeutic and biologic agents, and ET, may result in late-onset cardiotoxicity, increasing cardiovascular morbidity and mortality risk [82,83]. In addition, many chemotherapeutic agents used in the neoadjuvant setting are known for their potential to induce cardiac dysfunction [84]. Cancer survivors are 2–7 times more likely to die from cardiovascular diseases than the general population [85], with cardiac effects emerging as early as 6 months post-surgery or up to 8 years after diagnosis [44,45,46]. Long-term cardiac monitoring is essential, as subclinical damage can progress to heart failure years after treatment.

Anthracyclines, common chemotherapeutic agents, can cause cardiotoxicity, including acute or chronic cardiomyopathy, heart failure, arrhythmias, and even death. The risk increases with higher cumulative doses, affecting 1–11% of patients treated with epirubicin at 900 mg/m^2^ and 7–25% of patients treated with doxorubicin at 500–550 mg/m^2^, and requires close monitoring [86,87].

Patients with HER2+ eBC face additional cardiotoxicity risks from anti-HER2 drugs, including heart failure (2.5–4%) and decline in left ventricular ejection fraction (LVEF) (3–27%), meaning that cardiac monitoring is required before, during (every 3 months), and after (every 6 months for 2 years) treatment [88,89]. In patients with triple-negative eBC, immune checkpoint inhibitors (ICIs) rarely cause cardiotoxicity, but the appearance of severe cases of ICI-associated myocarditis and pericarditis highlights the need for thorough cardiac evaluation and continuous surveillance post-treatment [90].

The cardiovascular effects of tamoxifen and aromatase inhibitors (AIs) are still under investigation [91,92,93,94]. Tamoxifen, while potentially increasing the risk of stroke, has also demonstrated cardiovascular benefits, such as a lipid-lowering effect [92]. Current evidence suggests that tamoxifen may provide modest cardiovascular protection, particularly in postmenopausal women, regardless of pre-existing coronary heart disease [95]. In contrast, AIs have been associated with a higher risk of heart failure and cardiovascular mortality [51,96]. For example, a UK study of 17,922 patients reported that, compared to tamoxifen, AIs were linked to a 1.86-fold higher risk of heart failure and a 1.50-fold increase in cardiovascular mortality [97].

Anthracycline and taxane-based chemotherapy (ATAX) remains the standard of care for early-stage TNBC, with a lower heart failure risk compared to taxane-only regimens [98]. Radiotherapy, particularly for the left breast, can cause late cardiac toxicities, including coronary artery disease, acute myocardial infarction, cardiomyopathy, or valvular heart disease. However, modern radiotherapy techniques offer better cardiac protection [65,99].

Regular cardiac screening is essential for BC survivors who are at high risk of cardiotoxicity, as shown by the Cardiac-Related Oncologic Late Effects (CAROLE) study, where 77.6% of participants exhibited cardiovascular disease [66]. The study highlights the importance of adopting a multidisciplinary approach involving medical oncologists, radiation oncologists, cardiologists, and other healthcare providers to prevent and manage the late cardiotoxic effects associated with BC treatment. Strategies to reduce anthracycline cardiotoxicity include cardioprotective agents, modified protocols, and alternative therapies [64]. Personalizing treatment helps balance benefits and minimize risks.

### 3.3. Neurotoxicity

#### 3.3.1. Chemotherapy-Induced Peripheral Neuropathy (CIPN)

Adjuvant chemotherapy, especially with taxanes, can cause chemotherapy-induced peripheral neuropathy (CIPN), a common and potentially severe adverse effect of neoadjuvant chemotherapy (NACT). It is characterized by distal paresthesia (tingling and numbness), pain, and muscle weakness. CIPN prevalence ranges from 23 to 80% of patients [29,52,53], with a higher risk linked to cumulative doses (e.g., paclitaxel > 1000 mg/m^2^, docetaxel > 400 mg/m^2^), age, baseline neuropathy, smoking, and diabetes [67].

Persistent CIPN can lead to psychological distress, including depression and anxiety, and significantly affect HRQoL [100]. Regular assessment is crucial for the early detection of any issues. While there is no effective drug to prevent CIPN, cryotherapy, compression therapy, and medical exercise are options to consider. For chronic CIPN, duloxetine is the only treatment supported by level I evidence; other options include venlafaxine, pregabalin, amitriptyline, tramadol, or opioids. Acupuncture may also be considered in selected cases [101]. Currently, there are limited effective treatments for CIPN; once established, prevention and early detection are crucial for effective management. Emerging strategies, such as neuroprotective agents and prehabilitation programs, aim to reduce the incidence and impact of CIPN in NACT BC patients by optimizing their functional status before and during treatment [102].

#### 3.3.2. Cognitive Dysfunction

Cognitive dysfunction, or ‘chemobrain’, affects patients with BC who have undergone chemotherapy or ET, leading to issues with memory, attention, and executive function [54,68]. It impacts up to one in three patients and can persist for 2–3 years post-treatment [103].

The exact cause of chemobrain is unclear, but it may involve neurotoxic effects [104], inflammation, oxidative stress, and hormonal changes, with age and ET being risk factors [105,106]. Symptoms include trouble in multitasking, organizing, concentrating, forgetfulness, and reduced mental speed, sometimes impairing daily life and work [54]. Emotional distress, anxiety, and a reduction in overall HRQoL often accompany this condition. Depression, due to cognitive impairment, can also lead to decreased productivity and financial difficulties.

Interventions like cognitive rehabilitation, exercise, and low-evidence pharmacological treatments are being explored, but more research is needed to develop effective solutions [54].

### 3.4. Psychological Alterations

A diagnosis of BC is highly distressing for patients and their families, contributing to anxiety and depression due to the challenges of treatment, potential side effects, understanding the prognosis, and an uncertain future. This emotional distress can lead to a reduction in HRQoL and in treatment compliance while increasing mortality risk. Depression prevalence ranges from 9.4 to 66.1% among BC survivors, while anxiety affects 17.9–33.3% of patients [32].

Risk factors for anxiety include unemployment, younger age, physical symptoms, chemotherapy, poor social and cognitive functioning, and poor healthcare communication [55]. Depression is more strongly associated with a younger age at diagnosis, history of psychological disorder, substance abuse, poor social support, and lower socioeconomic status [69,107,108]. Fear of recurrence, affecting up to 71% of patients with BC, is particularly common in younger individuals [56] and in those undergoing breast-conserving surgery, compared to mastectomy [71].

To mitigate these effects, psychological support, psychiatric care, and cognitive–behavioral therapy are crucial for improving mental health outcomes during and after BC treatment [70,71].

### 3.5. Fatigue

Fatigue is a common symptom in patients with eBC, which can persist even after the completion of cancer treatment. The prevalence of fatigue as a long-term complication in these patients varies, but it is estimated to affect up to 30–50% of patients [33,43,56,109]. The cause of fatigue in BC survivors is multifactorial and can be related to either treatment or to other long-term adverse events such as cardiac, menopause, or psychological causes. Fatigue can have a significant negative impact on the HRQoL of BC survivors, affecting their ability to perform daily activities, work, and socialize. According to a meta-analysis of 12,327 patients, severe fatigue is more likely to occur in those at advanced stages of the disease, receiving chemotherapy, or undergoing a combination of surgery, RT, and chemotherapy with or without ET. Conversely, having family support or not receiving systemic treatment could reduce the risk of severe fatigue [72].

Management of fatigue can involve a combination of lifestyle modifications, such as regular exercise, adequate sleep, stress reduction techniques, and treatment of other comorbidities or late adverse events [110].

### 3.6. Hormonal Alterations

Patients on ET often experience persistent side effects, including genital atrophy, dizziness, weight gain, hot flashes, cognitive dysfunction, fatigue, and musculoskeletal impairment. These adverse effects can persist over a long period and have a negative impact on a patient’s HRQoL, leading to poor adherence to therapy [111]. For instance, hot flashes within 6 months of starting AIs increase the 5-year discontinuation rate by 14.2% [34]. Weight gain, affecting 50–96% of women with BC, is more pronounced in premenopausal women, those undergoing chemotherapy, and in those overweight at the time of diagnosis, potentially increasing mortality risk [112,113,114,115].

Studies have shown that ET has a significant negative impact on the HRQoL of patients with BC over time; symptoms such as joint pain, lack of energy, weight gain, and vaginal dryness persist in 33–48.7% of patients 5–10 years after diagnosis [40,57].

Nonhormonal pharmacotherapy, such as gabapentin or selective serotonin reuptake inhibitors/serotonin–norepinephrine reuptake inhibitors, may help women experiencing hot flashes. Though evidence is limited, homeopathy or herbal products are commonly used to mitigate hot flashes due to BC treatment [116]. Physical exercise, cognitive–behavioral therapy, and mindfulness have demonstrated efficacy in reducing menopausal symptoms in clinical trials [117,118,119].

### 3.7. Sexual Disorders and Fertility

Patients with BC have a high prevalence of sexual disorders (affecting up to 90% of patients) [58,120]. The main causes are associated with body image alterations, ET, and psychological impairment, such as depression or anxiety [36]. Treatment focuses on identifying and addressing the physical and/or psychological causes, rather than merely coping with the primary affliction. For example, in cases of vaginal dryness or dyspareunia, advising on the use of lubricants, or providing psychological support and sexual counseling in cases with depression or anxiety [73].

Young BC survivors may experience infertility due to chemotherapy-related gonadotoxicity and the delay in childbearing that is required when women are taking ET. This is an important concern for women, with one study showing that 36% of patients with eBC reported an interest in having children in the future [74]. In a meta-analysis of 112,840 patients with BC, only 6.5% became pregnant after diagnosis, and survivors had a 60% reduction in the likelihood of having a subsequent pregnancy compared with the general population [59]. Encouraging results were provided from the POSITIVE trial, involving 516 women aged 42 or younger and designed to evaluate the safety of temporarily pausing ET for up to 2 years, for women with HR+ eBC who wished to conceive. The study concluded that temporarily stopping ET for pregnancy did not significantly increase the short-term risk of BC recurrence. However, a longer follow-up is needed to confirm the long-term safety of this approach [121]. Oncofertility counseling should be offered to all women of reproductive age. Oocyte/embryo or ovarian tissue cryopreservation is the primary approach available for preserving fertility in patients with BC [122].

### 3.8. Gastrointestinal Symptoms

Gastrointestinal symptoms are not commonly reported as a long-term adverse effect, but rather as an early-onset event related to chemotherapy. This may change with the recent approval of new drugs to treat eBC, such as abemaciclib, which can cause diarrhea, or immunotherapies, like pembrolizumab. In the KEYNOTE-522 trial, 29.4% of patients treated with pembrolizumab reported some degree of diarrhea, and 62.7% reported nausea [123]. Published follow-up data are limited, and a longer follow-up period is needed to determine whether these symptoms persist in the long term.

In the MonarchE trial, in which patients received 2 years of adjuvant treatment with abemaciclib plus ET [30], 83% of patients experienced diarrhea that improved over time. Nonetheless, between 18 and 24 months during treatment, around 50% of patients still reported some degree of diarrhea. Nausea and vomiting were also reported (in 29.5% and 17.6% of patients, respectively). Further studies are needed to determine whether these symptoms persist post-treatment completion (2 years). In addition, the incorporation of immunotherapy in the early stages of BC treatment has made gastrointestinal symptoms more relevant. Nausea (28–77%), vomiting (40%), constipation (28–33%), and diarrhea (25–34%) are also frequently reported in patients with BC receiving olaparib treatment [124]. Most recently, in the NATALEE trial reporting the benefits of ribociclib in patients with eBC, 23% patients experienced some grade of nausea [17]. Overall, more follow-up is needed to determine whether these symptoms persist in the long term.

### 3.9. Endocrine Toxicity

ICIs, HT, and radiotherapy lead to dysregulation of immune homeostasis and endocrine toxicity, mainly as thyroid dysfunction, and diabetes mellitus [125].

Radiotherapy is associated with an increased risk of hypothyroidism [126]. In eBC, hypothyroidism was registered in 5.7% of patients at a median time of 3.45 years from the index date [35], with no association with the systemic oncological treatment. Digkas et al. identified a 68% higher risk for hypothyroidism in patients treated with radiotherapy of the regional lymph nodes, with no association observed between hypothyroidism and chemotherapy, ET, or radiotherapy to the breast/chest wall, irrespective of the use of adjuvant chemotherapy treatment. As for endocrine toxicity due to HT, several studies have suggested an association between eBC treatment and diabetes mellitus. Ye et al. recently reported a 30% higher risk of diabetes for patients with primary BC after HT (hazard ratio 1.30), compared to non-HT treated patients, and identified a 15% incidence rate of diabetes in HT treated patients with BC [127].

Endocrine dysfunction appears as one of the most common adverse events after ICI across multiple solid tumors [128] and an evaluation of immune-related adverse events after pembrolizumab treatment reported hypothyroidism and hyperthyroidism in 20% and 2.9% of patients, respectively [129], revealing a relatively high incidence of thyroid dysfunction during eBC treatment.

### 3.10. Osteomuscular Alterations

Patients with eBC treated with adjuvant therapy face an increased risk of bone loss and osteoporosis. HT significantly disrupts skeletal metabolism, making bone integrity a major concern [130]. AIs reduce estrogenic levels, critical for maintaining bone strength, leading to bone density loss and fracture risk, particularly in postmenopausal women [131,132,133]. Tamoxifen protects bones in postmenopausal women but may cause bone loss in premenopausal women once discontinued [134].

Muscle alterations, including sarcopenia, are another concern. Chemotherapy disrupts muscle protein balance, reducing muscle mass and strength, affecting their overall HRQoL and increasing the risk of other toxicities [135]. Extended HT use further increases the risk of fractures, osteoporosis, bone pain, myalgias, and treatment discontinuation. Physical inactivity, often driven by treatment fatigue, worsens muscle health [136].

Addressing these issues requires interventions such as weight-bearing exercises, calcium and vitamin D supplementation, and pharmacological treatments (e.g., bisphosphonates, denosumab), along with regular monitoring, to support long-term recovery.

## 4. Patients with BRCA-Mutated Tumors

Approximately 10% of all BC cases are due to germline mutations in the *BRCA1* and *BRCA2* genes (gBRCAm), and these patients are at an increased risk of developing a second BC. The toxicity profile of anticancer drugs in these patients may differ from non-gBRCAm patients, as they are on average younger at diagnosis and usually undergo more aggressive treatment approaches, including bilateral mastectomy as the first local treatment recommendation [137] and the addition of bilateral oophorectomy (inducing early menopause) to decrease the risk of a second BC (or primary ovarian cancer). Importantly, recent large-scale international data have confirmed a survival benefit associated with these risk-reducing surgeries in young BRCA patients [138]

Despite these features, treatment consequences in this subgroup of patients are not well studied, as most studies focus on early adverse events, and data for late-onset effects are scarce [139]. While our understanding of long-term adverse events in gBRCAm patients is still evolving, it is crucial to define and characterize the potential impact of early-onset adverse events on patients’ long-term health outcomes.

It is not well established if patients with gBRCAm BC are more likely to develop adverse events post-treatment than those with non-gBRCAm BC. Some studies have shown that patients harboring a gBRCAm BC tumor could present with hematological adverse effects earlier. In a retrospective analysis that included 270 patients, those harboring a gBRCAm appeared to have a higher risk of developing neutropenia after initial anthracycline-based chemotherapy [140], while initial trials with PARP inhibitors suggest higher rates of anemia, lower white-cells counts, fatigue, and lymphopenia post-treatment [18]. Interestingly, recent evidence indicates that there might be a slightly higher risk of contralateral BC in irradiated gBRCA1/2m patients [141]. This study included 3602 patients from the International BRCA1/2 Carrier Cohort Study (IBCSS) and explored the relationship between radiation therapy for primary BC and the risk of developing contralateral BC in gBRCA1/2m patients. Conversely, non-gBRCAm patients showed a significant increase in nausea and vomiting.

In another retrospective analysis, the rate of severe hematological toxicities in patients treated with taxanes was higher in those with gBRCAm compared with the non-gBRCAm control group (59.5% vs. 43.1%; *p* < 0.001) [142]. In contrast, a larger retrospective analysis showed that anemia and leukopenia were more frequently seen in patients with non-gBRCAm who were receiving taxane-containing chemotherapy compared with patients with gBRCAm [143,144]. The effect of radiotherapy toxicity has been examined in several studies, including patients with gBRCAm, and it was found that rates of radiation-induced adverse events were similar to those in women with sporadic BC. In their matched case–control study, Shanley et al. found no significant differences in acute and late radiation effects between patients with gBRCAm (*n* = 55) and those with sporadic BC (*n* = 55) [145]. Similarly, Pierce et al. compared the rates of chronic skin, subcutaneous tissue, lung, and bone adverse events in patients with stage I or II BC treated with breast-conserving surgery and radiotherapy. They found no significant differences between the genetic (*n* = 71) and sporadic cohorts (*n* = 213) [146]. Moreover, Park et al. observed no increased risk of acute skin toxicity in non-Caucasian patients with gBRCAm (*n* = 46) who underwent breast-conserving therapy using radiotherapy compared to women with sporadic BC [147]. More recently, Vliek et al. identified fatigue, nausea, and infections as the most common side effects after high-dose alkylating chemotherapy vs. standard neoadjuvant therapy in TNBC gBRCAm patients, as reported in the randomized phase 3 NeoTN trial [148].

Most studies on long-term adverse events in gBRCAm patients have focused particularly on cardiotoxicity, gonadotoxicity, and the psychological impact of the disease. A retrospective study of 898 patients (167 gBRCA1m, 91 gBRCA2m, and 640 non-gBRCAm) found no difference in hematological events, cardiac alterations, or neuropathy [149]. Also, gBRCAm was not associated with an increased risk of peripheral neuropathy [143]. In terms of cardiotoxicity, it seems that gBRCAm does not confer a higher risk of toxicity. In a single exploratory study involving 401 patients, comprising 232 *BRCA1* and 159 *BRCA2* mutation carriers, the authors reported a higher likelihood of heart failure based on self-reported symptoms obtained from an anonymous survey compared to historical controls from the general population. However, this study lacked objective confirmatory data, such as echocardiogram reports for most participants, and did not have a direct comparator control group [150]. Furthermore, two other studies found no significant differences in the rates of cardiomyopathy between gBRCAm carriers and non-carrier controls who received anthracyclines [151,152]. In a separate cross-sectional study, 67 patients with an average time of 6 years since BC diagnosis were enrolled. This study showed that women with gBRCAm did not have an increased risk of anthracycline-induced cardiotoxicity compared to those with sporadic BC [153].

Despite the increasing amount of available data on the safety and feasibility of preserving fertility and achieving pregnancy after BC diagnosis in the general population, there are still significant challenges facing patients with gBRCAm. Limited research has been conducted in this group, making it particularly difficult to provide accurate counseling on the risk of chemotherapy-induced gonadotoxicity [154]. In a multicenter survey conducted by Valentini et al., 1954 premenopausal gBRCA patients were studied, 1426 of whom received chemotherapy. Notably, the risk of premature ovarian failure was significantly higher in *BRCA2* carriers than in *BRCA1* carriers (46.8% vs. 32.7%; *p* < 0.001), but there was no significant difference in risk between non-carrier controls and either *BRCA1* carriers or *BRCA2* carriers [155]. Further research is needed to investigate the impact of treatment on the fertility of these patients and the effectiveness of fertility preservation techniques in this population.

Psychological impairment in patients with gBRCAm is also a crucial topic that deserves attention. BC diagnosis is a significant stressor for patients that can be exacerbated after gBRCAm diagnosis. A systematic review of eight studies that investigated the psychological impact of gBRCAm testing found that there was a negative effect on psychological well-being in the first months after positive test disclosure, with increased symptoms of distress, anxiety, and depression. In contrast, no significant clinical symptoms were observed in the intermediate and long-term periods. However, the lack of specific measurements that can accurately identify the psychological burdens of cancer-affected mutation carriers highlights the need for more research in this area [156]. Thus, further studies should focus on developing more precise and reliable screening tools to identify the long-term psychological impact of a positive gBRCAm diagnosis disclosure, allowing for better support and care for those affected women. Understanding the psychological impact of gBRCAm on patients can help healthcare providers to provide better support, reduce distress, and improve the overall HRQoL.

In summary, the available evidence suggests that patients with gBRCAm may have different patterns of toxicity during BC treatment, but the overall incidence is not significantly increased in these patients (Figure 2). However, further research is needed to clarify the association between gBRCAm and long-term adverse events, particularly in the context of novel targeted therapies and immunotherapies.

## 5. Current Research Needs in Breast Cancer Survivorship

Increasing BC survival rates due to advances in diagnosis and treatment have led to a growing population of BC survivors experiencing the physical and psychological side effects of treatment. Current European Society For Medical Oncology guidelines emphasize monitoring BC survivors for long-term psychological and neurobiological disturbances, including anxiety, depression, sleep disturbances, chronic fatigue, and neurocognitive dysfunction, among others [22].

In 2016, experts identified key priorities to improve BC management, including therapy de-escalation, optimizing the duration of adjuvant treatment, improving care for young patients, and focusing on survivorship and HRQoL [39]. These efforts underscore the growing emphasis on both the physical and psychological aspects of BC patient care.

Emerging treatments, such as immunotherapy, antibody–drug conjugates, and CDK4/6 inhibitors, present new opportunities but also challenges, including the potential for long-term adverse effects (e.g., endocrinopathies, arthritis, xerostomia, neurotoxicity, neutropenia, and gastrointestinal symptoms). While these toxicities are well-studied in metastatic disease, their long-term impact in eBC remains unclear [125,157,158]. Recent trials, like PEONY (phase 3) and coopERA (phase 2), highlight adverse events such as neutropenia, leukopenia, and low white blood cell numbers as emerging adverse factors that require further evaluation [159,160]. Clinical data from the KEYNOTE-522 trial include immune-related adverse events (including hypothyroidism, arthritis, hepatitis, and dermatitis, among others) in a significant subset of patients, linked to treatment response [161], underscoring the need for monitoring as these therapies gain wider use. Additionally, patients with gBRCAm eBC treated with a PARPi, such as olaparib, in the OlympiA clinical trial [18] may face unique adverse effects, including pneumonitis and new primary cancers, with further research needed to understand and manage the long-term impact.

## 6. Gaps and Future Directions in the Study of Long-Term Adverse Events in BC Survivors

Our review of the current literature reveals variations and gaps in the current understanding of long-term adverse events in BC (see Appendix A for a summary of the studies cited). Despite significant advances in documenting treatment-related complications, several limitations persist that warrant further exploration.

Limited integration across affected systems: most studies examine individual domains, such as cardiotoxicity [44,66], CIPN [29,67], or psychological distress [32,55], in isolation. There is a lack of comprehensive frameworks to assess how these symptoms may interact over time and to determine their combined impact on the overall quality of life of BC survivors.Inadequate stratification by patient subgroups: while gBRCA1/2 carriers and younger women have been studied in specific contexts [143,156], many studies fail to stratify findings by age, menopausal status, race/ethnicity, or comorbidities. This limits the ability to deliver tailored survivorship care that addresses the distinct risks and experiences of diverse BC patient populations.Understudied and emerging adverse events: several important domains remain under-researched in long-term BC survivorship. These include endocrine dysfunction [35,126], sexual health and intimacy issues [36,58], weight gain, and metabolic syndrome [112,113]. These outcomes are rarely included in clinical trials and are inconsistently addressed in follow-up care plans, limiting comprehensive survivorship planning.Inconsistency in the definition of long-term adverse events, including time frames: there is considerable variability in how studies define long-term and late adverse effects. Some studies classify them as events persisting beyond 1 year post-treatment, while others only focus on those emerging after 5 years [38,42]. Additionally, the heterogeneity in study endpoints, from clinical indicators to patient-reported outcomes, hampers cross-study comparisons and limits the ability to synthesize findings effectively.Need for longitudinal, survivorship-focused research: some clinical trials emphasize efficacy (e.g., disease-free survival) over survivorship quality metrics [16,123]. Long-term safety and well-being outcomes require more deliberate inclusion in research study designs.Opportunity for retrospective studies using electronic health records (EHRs): retrospective analysis of large EHR databases offers a promising avenue to study the impact of long-term adverse events on BC survivorship at scale in real-world clinical practice, beyond a controlled clinical trial setting. Several studies have demonstrated the utility of advanced natural language processing and machine learning to extract and analyze longitudinal data from unstructured clinical notes, enhancing the detection of late toxicities and comorbidities [162,163,164,165]. Leveraging such EHR-based platforms can fill critical gaps, especially in understudied patient subpopulations and rare adverse events, while providing insights into BC survivorship care patterns and outcomes across diverse healthcare systems.

## 7. Conclusions

As advances in early detection and treatment have improved survival rates for eBC, it has become increasingly essential to prioritize the management of long-term adverse events associated with treatment. These cumulative adverse effects can profoundly impact patients’ HRQoL, overall health, and sustained well-being beyond the initial treatment period.

This review highlights that while improving survival remains a primary goal, equal attention must be given to developing less toxic therapeutic options and employing personalized medicine approaches that tailor treatment to the individual patient’s risk profile. This personalized approach is crucial for mitigating long-term toxicities and optimizing patient outcomes. These treatment plans will provide a framework for continuous monitoring and supportive care, addressing patients’ evolving needs throughout their survivorship journey.

Although the subpopulation of patients with gBRCAm may experience added psychological stress and distinct clinical challenges during treatment, the current evidence does not conclusively demonstrate that their long-term adverse events differ substantially from those experienced by patients with non-gBRCAm BC. Nevertheless, these individuals require particular attention in survivorship programs to support their unique psychosocial and medical needs.

Looking ahead, future research must focus on integrated patient-centered survivorship frameworks with standardized definitions of adverse events, the inclusion of underrepresented BC patient populations, and extended follow-up periods. Such efforts will be vital to understanding and managing the long-term sequelae of BC treatment fully. Moreover, evidence derived from real-world clinical practice and EHRs will offer valuable insights that can inform individualized care strategies. By leveraging these data, healthcare providers can better align interventions with patient-specific needs, ultimately enhancing patient satisfaction and quality of care.

In summary, this review highlights the need to shift the focus from survival alone to a more comprehensive approach to BC care that incorporates the long-term management of adverse events and prioritizes HRQoL. This paradigm shift is essential to ensure that extended survival is matched by a life of sustained health, dignity, and well-being for BC survivors.

## Figures and Tables

**Figure 1 cancers-17-02506-f001:**
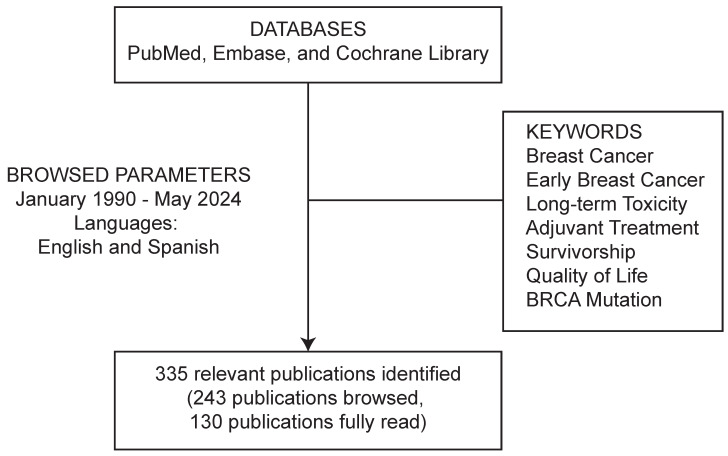
Literature search and review approach.

**Figure 2 cancers-17-02506-f002:**
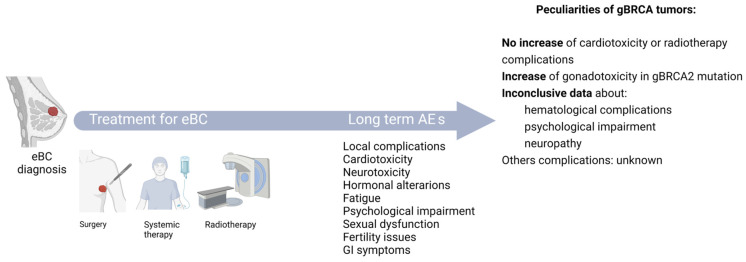
Schematic representation of the possible early and later-onset adverse events related to treatment in patients with eBC and gBRCA mutation. AE: adverse event, eBC: early breast cancer, gBRCA: germline BRCA, GI: gastrointestinal. Red dot represents tumor location.

**Table 1 cancers-17-02506-t001:** Time period considerations for BC adverse events.

Long-Term Adverse Events	Time Span
Chest wall and breast adverse events	
PMPS	7–12 years after BC diagnosis [47]
Lymphedema	10 years after BC diagnosis [48] After BC primary treatment [49]
Skin and soft tissue affection	After BC primary treatment [50]
Cardiologic	
Heart failure	>6 months after BC diagnosis [51], 5–8 years after BC diagnosis [45], 8 years after BC diagnosis [45], 10–13 years after BC diagnosis [46], and 11 years after BC diagnosis [44]
Arrhythmia, acute ischemic heart disease, ischemic stroke, or transient ischemic attack	>6 months after BC diagnosis [51]
Neurotoxicity	
CIPN	>3 weeks after BC treatment [52], after the first administration of chemotherapy [53]
Cognitive dysfunction	During chemotherapy treatment, after cessation of treatment, >6 months post-treatment cessation, >1 year post-treatment cessation, >3 years post-treatment cessation [54]
Psychological alterations	
Anxiety	Different timepoints ranging from 1.8 to 21 years after BC diagnosis [55]
Depression	Different timepoints ranging from 1.8 to 21 years after BC diagnosis [55]
Fear of death	>1 year after BC diagnosis [56]
Women’s health	
Fatigue	After BC primary treatment, >5 years after BC diagnosis [43]
Hormonal alterations	After BC primary treatment [40], >8 years after BC diagnosis [57]
Sexual disorders	>3 years after BC diagnosis [58]
Reduced fertility	>2 years after BC diagnosis [59]
GI symptoms	
Diarrhea	After BC primary treatment [30]
Endocrine symptoms	
Hypothyroidism	>3 years after BC diagnosis [35]
Osteomuscular adverse events	
Osteoporosis	After BC primary treatment [60]

BC: breast cancer, CIPN: chemotherapy-induced peripheral neuropathy, GI: gastrointestinal, PMPS: post-mastectomy pain syndrome. The gray background indicates different categories of adverse events.

**Table 2 cancers-17-02506-t002:** Description of key adverse events in patients with BC: prevalence, associated risk factors, and management.

Type of Adverse Event	Prevalence	Risk Factors	Management
Chest wall and breast			
PMPS [61,62]	28.2–65%	Postoperative pain, younger age, high BMI, axillary radiation, and axillary lymph node dissection	Analgesics, surgical interventions, acupuncture, or hypnosis
Lymphedema [28,48,49]	27–40%	ALND, mastectomy, adjuvant therapies, high BMI	Physiotherapy
Skin and soft tissue affections [31,63]	Up to 43%	Radiotherapy	Physiotherapy, anti-inflammatory drugs
Cardiologic			
Cardiac toxicity [64,65,66]	1–51.5%	Age, history of heart disease, maximum cumulative dose of anthracyclines, endocrine therapy, radiation to the left breast	Prevention: use of alternative chemotherapeutic agents, cardioprotective agents Treatment: same guidelines for heart failure for other causes
Neurologic			
CIPN [29,67]	23–80%	Age, taxane treatment, baseline neuropathy, smoking, diabetes	Duloxetine (level I evidence), venlafaxine, pregabalin, amitriptyline, and tramadol. In selected patients, acupuncture can also be an option
Cognitive dysfunction [54,68]	28–33%	Age, chemotherapy, endocrine therapy	Cognitive rehabilitation, physical exercise, and low evidence for pharmacological treatment
Psychological alterations			
Depression [32,69]	9.4–66.1%	Younger age at diagnosis, history of psychological disorder, substance abuse, poor social support, and lower socioeconomic status.	Psychological/psychiatric support and cognitive–behavioral therapy
Anxiety [32,70]	17.9–33.3%	Younger age, physical symptoms, chemotherapy, poor social and cognitive functioning, and communication problems with healthcare providers	Psychological/psychiatric support and cognitive–behavioral therapy
Fear of death [56,71]	71%	Uncertain future, young age, breast-conserving surgery	Psychological/psychiatric support and cognitive–behavioral therapy
Women’s health			
Fatigue [33,72]	30–50%	Relation with long-term adverse events such as cardiac, menopause, or psychological	Lifestyle modifications, such as regular exercise, adequate sleep, stress reduction techniques, and treatment of other comorbidities or late adverse events
Hormonal alterations [34,40]	33–48.7%	Endocrine therapy, chemotherapy	Gabapentin or SSRIs/SNRIs for hot flashes. Physical exercise, cognitive-behavioral therapy, and mindfulness
Sexual disorders [58,73]	90%	Body image alterations, endocrine therapy, and psychological impairment, such as depression or anxiety	The treatment of associated factors (vaginal dryness, dyspareunia, depression, or anxiety, etc.) Sexual counseling
Reduced fertility [59,74]	60%	Gonadotoxic chemotherapy	Oncofertility counseling
GI symptoms			
Diarrhea [30]	29.4–83%	Treatment with abemaciclib or immunotherapy	Dose reduction or interruption according to severity. Loperamide for abemaciclib toxicity and corticosteroids for immunotherapy according to severity
Nausea [30]	23.0–77%	Treatment with abemaciclib, olaparib, or ribociclib	Integration of strategies to prevent or lessen its impact
Vomiting [30]	40%	Treatment with olaparib	Integration of strategies to prevent or lessen its impact
Endocrine symptoms			
Hypothyroidism [35]	5–6%	Radiotherapy treatment	Hormonal supplementation

ALND: axillary lymph node dissection, BMI: body mass index, CIPN: chemotherapy-induced peripheral neuropathy, GI: gastrointestinal, PMPS: post-mastectomy pain syndrome, SSRIs/SNRIs: selective serotonin reuptake inhibitors/serotonin–norepinephrine reuptake inhibitors. The gray background indicates different categories of adverse events.

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
