# Peer review of "Long-Term Adverse Events Following Early Breast Cancer Treatment with a Focus on the BRCA-Mutated Population"

_cancers, 2025, doi:10.3390/cancers17152506_

Round 1
Reviewer 1 Report
Comments and Suggestions for Authors
This manuscript is a review article focused on Long-Term Adverse Events Following Early Breast Cancer Treatment. It is divided into two parts. The first part, which is the longer analysis, examines the long-term adverse events in patients with early breast cancer. The second part addresses these events in breast cancer cases associated with germline mutations in the BRCA1 and BRCA2 genes (gBRCAm). The authors have put in considerable effort by reviewing around 150 articles from scientific databases worldwide. The paper is well-written, and the facts and explanations are clear and accessible. There are evident connections between different sections of the manuscript.
As the authors note, there are no standardized definitions for what constitutes long-term adverse events in breast cancer patients. This lack of standardization is due to various factors, including differing treatments, external influences, and genetic predispositions. To address this, they have classified the events by symptoms.
I have no significant objections to the structure or content of the manuscript. The authors have presented this review article in a narrative form, as they describe.
Is it possible to create additional classifications besides those on symptoms?
Are there differences or gaps among some of the studies included in this review article? Perhaps some open issues could be highlighted or discussed in a separate mini-section.
It would also be beneficial to include references in Table 2, similar to those given in Table 1.
This paper merits publication as a review article in this journal, pending minor revisions.
Author Response
Comments and Suggestions for Authors
This manuscript is a review article focused on Long-Term Adverse Events Following Early Breast Cancer Treatment. It is divided into two parts. The first part, which is the longer analysis, examines the long-term adverse events in patients with early breast cancer. The second part addresses these events in breast cancer cases associated with germline mutations in the BRCA1 and BRCA2 genes (gBRCAm). The authors have put in considerable effort by reviewing around 150 articles from scientific databases worldwide. The paper is well-written, and the facts and explanations are clear and accessible. There are evident connections between different sections of the manuscript.
As the authors note, there are no standardized definitions for what constitutes long-term adverse events in breast cancer patients. This lack of standardization is due to various factors, including differing treatments, external influences, and genetic predispositions. To address this, they have classified the events by symptoms.
I have no significant objections to the structure or content of the manuscript. The authors have presented this review article in a narrative form, as they describe.
Thank you very much for your thorough and thoughtful review of our manuscript. We appreciate your positive feedback regarding the clarity, structure, and comprehensive nature of our review. We are pleased that the organization of the manuscript and the connections between sections were found to be coherent and accessible.
We also acknowledge your recognition of the challenges posed by the lack of the standardized definitions for long-term adverse events in breast cancer (BC) patients, and we agree that classifying these events by symptoms was a practical approach to address this issue.
Your supportive comments encourage us to believe that the narrative format we chose effectively conveys the multifaceted nature of long-term adverse events following early BC treatment, including important considerations related to germline BRCA mutations. We hope this work will contribute meaningfully to the field and aid clinicians and researchers in understanding and managing these long-term effects.
Is it possible to create additional classifications besides those on symptoms?
Thank you for your insightful question regarding the possibility of additional classifications beyond symptom-based frameworks. Certainly, it is definitely possible, and often helpful, to classify long-term effects of BC patients using frameworks beyond just symptoms. While symptom-based classifications are important for managing daily experiences, additional classifications can provide a broader understanding of survivorship needs and help guide more personalized care. Long-term effect could be classified according to the type of impact on life (i.e., physical functioning, cognitive functioning, emotional and psychological well-being, social and occupational impact, or sexual reproductive health), by the treatment modality received (i.e., surgery-related, chemotherapy-related, radiation-related, hormonal therapy-related, or targeted therapy-related), by symptoms severity (i.e., mild, moderate, or severe), or by patient-reported outcomes (i.e., health-related quality of life, emotional distress, fatigue burden, life satisfaction, or coping and resilience). However, we identified a gap in literature concerning long-term effects on specific body systems, which is of significant interest to the medical community because it may facilitate clearer referral pathways, improve interdisciplinary collaboration, and support more structured follow-up protocols for BC survivors. We hope this review contributes to addressing that need and supports healthcare professionals, both doctors and nurses, in their clinical practice. We have incorporated this important reflection into the revised manuscript.
Are there differences or gaps among some of the studies included in this review article? Perhaps some open issues could be highlighted or discussed in a separate mini-section.
We thank the reviewer for this suggestion. In response, we have carefully revisited the studies included in the review and agree that highlighting differences and gaps among them adds important context and strengthens the manuscript.
As a result, we have added a new mini-section titled “Gaps and Future Directions in the Study on Long-Term Adverse Events in Breast Cancer Survivors”. This section synthesizes key observations regarding the variability in study focus, limited stratification by demographic and clinical subgroups, underexplored areas such as endocrine dysfunction, metabolic complications, and sexual health, inconsistent definition of “long-term” and “late” effects, and the need for prospective, survivorship-focused studies with standardized outcome measures.
By incorporating this section, we aim to provide a more nuanced and forward-looking view of the current literature, and to identify open research questions that remain to be addressed in the field.
It would also be beneficial to include references in Table 2, similar to those given in Table 1.
Thank you for this helpful suggestion. We have revised Table 2 to include appropriate references for each of the listed adverse events, following the same format as used in Table 1. These citations correspond to key studies discussed in the main text and provide additional context for the evidence supporting each item listed. We believe this enhancement improves the transparency and scholarly value of the table, and we appreciate the reviewer’s recommendation.
This paper merits publication as a review article in this journal, pending minor revisions.
We thank again the reviewer for their positive evaluation of our manuscript and for recommending it for publication as a review article, pending minor revisions. We greatly appreciate the time and expertise invested in assessing our work.
In response to the comments provided, we have carefully addressed all suggested minor revisions to enhance the clarity, accuracy, and overall quality of the manuscript. Detailed responses to each point are provided above, along with a revised version of the manuscript highlighting the changes made.
We are grateful for the reviewer’s constructive feedback and support, and we hope that the revised version meets the expectations for publication.

Reviewer 2 Report
Comments and Suggestions for Authors
Manuscript ID: 3722697
Manuscript Title: “Long-Term Adverse Events Following Early Breast Cancer Treatment with a Focus on the BRCA Mutated Population”
Thank you for the opportunity to review this manuscript. The study aims to examine long-term adverse events, their associated risk factors, and management strategies, with a particular focus on patients with germline BRCA mutations (gBRCAm). The manuscript falls within the scope of the journal and addresses an original and relevant topic.
However, there are several issues that need to be addressed:
-
Line 96: Please include the percentage of patients experiencing gastrointestinal symptoms.
-
Table 1: Add a legend to clarify all acronyms used in the table.
-
Paragraph 3.1.1: It should be stated that in recent years there has been a clear trend toward surgical de-escalation, with the increased use of conservative mastectomies. The reported 72% mastectomy rate appears excessively high—this should be reformulated.
-
Paragraph 3.1.2: The same observation as above applies. This paragraph should be revised to reflect current practice. Please also update the references, as the ones currently cited date back to 2008.
-
Chapter 4: Please insert the following reference:
Blondeaux E, et al. Association between risk-reducing surgeries and survival in young BRCA carriers with breast cancer: an international cohort study. Lancet Oncol. 2025 Jun;26(6):759-770. doi: 10.1016/S1470-2045(25)00152-4. Epub 2025 May 8. PMID: 40347973. -
End of the Discussion: A summary table of the studies cited should be added, indicating the author, year, study design (prospective or retrospective), number of patients, and the long-term effects considered.
-
Neoadjuvant Chemotherapy (NACT): Please include more specific information about patients who underwent NACT. Are there any long-term outcomes described specifically for this subgroup? We suggest citing:
Di Leone A, et al. Neoadjuvant Chemotherapy in Breast Cancer: An Advanced Personalized Multidisciplinary Prehabilitation Model (APMP-M) to Optimize Outcomes. J Pers Med. 2021 Apr 21;11(5):324. doi: 10.3390/jpm11050324. -
Conclusions: The conclusion section should be reformulated more clearly. What are the key takeaways from this review? Can it be definitively stated that long-term adverse effects differ in BRCA-mutated patients?
At this stage, the manuscript cannot be considered eligible for publication and requires major revisions.
Author Response
Comments and Suggestions for Authors
Manuscript ID: 3722697
Manuscript Title: “Long-Term Adverse Events Following Early Breast Cancer Treatment with a Focus on the BRCA Mutated Population”
Thank you for the opportunity to review this manuscript. The study aims to examine long-term adverse events, their associated risk factors, and management strategies, with a particular focus on patients with germline BRCA mutations (gBRCAm). The manuscript falls within the scope of the journal and addresses an original and relevant topic.
However, there are several issues that need to be addressed:
Line 96: Please include the percentage of patients experiencing gastrointestinal symptoms.
Thank you for your suggestion. We have revised the manuscript to include the percentage of patients experiencing gastrointestinal (GI) symptoms, as reported in specified reference. We include the data from the MonarchE trial, which found that 42.8% of patients experiences grade ≥2 diarrhoea, the most reported gastrointestinal adverse event. This information has been added to the main text to provide a clearer picture of the burden of GI symptoms in BC treatment.
Table 1: Add a legend to clarify all acronyms used in the table.
Thank you for this helpful suggestion. We have now added a detailed legend to both tables that defines all acronyms and abbreviations used, ensuring clarity and ease of understanding for readers. This addition enhances the tables accessibility and improves the overall readability of the manuscript.
Paragraph 3.1.1: It should be stated that in recent years there has been a clear trend toward surgical de-escalation, with the increased use of conservative mastectomies. The reported 72% mastectomy rate appears excessively high—this should be reformulated.
Thank you for this important observation. We agree with your suggestion to highlight the recent trend toward a surgical de-escalation in BC treatment. Over the past decade, there has been a significant shift toward less invasive surgical approaches, including increased utilization of breast-conservative surgery and conservative mastectomies, aiming to reduce morbidity without compromising oncological outcomes. This evolving surgical landscape is important context for interpreting current and future rates of postoperative complications, including chronic pain.
In addition, we would like to clarify that the figure of 72% does not refer to the rate of mastectomy, but rather to the reported incidence of chronic or nerve damage following BC surgery, as noted in the referenced literature. This percentage reflects a broad range of pain prevalence (20-72%) depending on whether continuous or intermittent symptoms are included. To avoid any confusion, we have revised the sentence for clarity as follows:
The incidence of nerve damage and chronic pain following BC surgery has been reported to range from 20% to 72% depending on the definition and duration of symptoms. Risk factors for persistent pain include postoperative pain, younger age, high BMI, axillary radiation, and axillary lymph node dissection. It is important to note that in recent years, surgical de-escalation strategies, such as increased use of breast-conserving surgery and conservative mastectomies, have been widely adopted, which may influence the prevalence and severity of long-term AEs.
Paragraph 3.1.2: The same observation as above applies. This paragraph should be revised to reflect current practice. Please also update the references, as the ones currently cited date back to 2008.
We thank the reviewer for this valuable suggestion. In response, we have revised the paragraph to better reflect current clinical practice and have updated the supporting references accordingly. The revised text incorporates recent evidence on the prevalence, risk factors, and modern management strategies for lymphoedema, with emphasis on evidence-based, non-surgical interventions and the importance of early intervention.
Specifically, we have added more recent and comprehensive references, including Gao et al. (2023) and Shamsesfandabadi et al, (2025). All references included in this section provide updated epidemiological data and guideline-based recommendations on psychotherapy and conservative care approaches. We believe these revisions offer a more accurate representation of current standards in lymphoedema care and survivorship support.
Chapter 4: Please insert the following reference:
Blondeaux E, et al. Association between risk-reducing surgeries and survival in young BRCA carriers with breast cancer: an international cohort study. Lancet Oncol. 2025 Jun;26(6):759-770. doi: 10.1016/S1470-2045(25)00152-4. Epub 2025 May 8. PMID: 40347973.
We thank the reviewer for this valuable suggestion. We have reviewed the study by Blodeaux et al. (2025) and agree that it is highly relevant to the statements described in Chapter 4 regarding BRCA carriers and risk-reducing surgical strategies. We have now incorporated this reference into the chapter to strengthen the evidence base and provide an updated perspective on survival outcomes in this patient population. The first paragraph of Chapter 4 has been updated and revised adding the following text:
Importantly, recent large-scale international data have confirmed a survival benefit associated with these risk-reducing surgeries in young BRCA patients (Blondeaux et al., 2025)
End of the Discussion: A summary table of the studies cited should be added, indicating the author, year, study design (prospective or retrospective), number of patients, and the long-term effects considered.
Thank you for your valuable suggestion regarding the inclusion of a summary table at the end of the Discussion section. We agree that such a table would greatly enhance the manuscript by providing readers with a clear and concise overview of the key studies referenced throughout the review. In response, we have added a comprehensive summary table (Supplementary Table 1) that includes the author names (with numbered reference), publication year, study design (prospective or retrospective), number of patients, and the specific long-term adverse events evaluated in each study. This addition aims to improve the accessibility and usability of the information presented, allowing readers to better appreciate the scope and strength of the existing evidence. We believe this enhancement will complement the narrative discussion and support a more informed understanding of the long-term outcomes following early BC treatment.
Neoadjuvant Chemotherapy (NACT): Please include more specific information about patients who underwent NACT. Are there any long-term outcomes described specifically for this subgroup? We suggest citing:
Di Leone A, et al. Neoadjuvant Chemotherapy in Breast Cancer: An Advanced Personalized Multidisciplinary Prehabilitation Model (APMP-M) to Optimize Outcomes. J Pers Med. 2021 Apr 21;11(5):324. doi: 10.3390/jpm11050324.
We appreciate the reviewer’s suggestion to provide more specific information regarding patients who underwent neoadjuvant chemotherapy (NACT) and to discuss long-term outcomes specific to this subgroup. In response, we have expanded the manuscript to include detailed descriptions of adverse events observed in patients receiving NACT. Rather than creating a separate section exclusively for NACT patients, we have integrated these particular adverse events into the pre-existing sections, categorizing them according to the type of event. This approach allows a more cohesive and comprehensive presentation of treatment-related toxicities across the entire cohort of BC patients while highlighting those relevant to NACT patients.
Furthermore, we have enriched the discussion with recent and pertinent references that specifically address long-term outcomes for patients underdoing NACT. These additions provide a more nuanced understanding of the long-term implications of NACT and reinforce the importance of tailored management to optimize patient outcomes.
We trust these revisions satisfactorily address the reviewer’s concerns
Conclusions: The conclusion section should be reformulated more clearly. What are the key takeaways from this review? Can it be definitively stated that long-term adverse effects differ in BRCA-mutated patients?
We thank the reviewer for this valuable comment regarding the clarity and focus of the conclusion section. In response, we have thoroughly revised the conclusion to more clearly articulate the key takeaways from the review. The updated version now explicitly highlights the importance of shifting the focus from survival alone to long-term management of adverse events, emphasizing the role of personalized medicine, survivorship care planning, and real-world data in improving patient outcomes and quality of life.
Additionally, we have addressed the reviewer’s question regarding BRCA-mutated (gBRCAm) patients. Based on the available evidence, we acknowledge that while these patients may experience unique psychosocial and clinical challenges during treatment, there is currently no definitive evidence to suggest that the long-term adverse effects they experience differ substantially from those observed in non-gBRCAm patients. This has been explicitly stated in the revised conclusion to avoid overgeneralization and maintain scientific accuracy.
We hope that the revised conclusion now more clearly conveys the key messages of the review and appropriately reflects the current state of evidence regarding long-term adverse effects in gBRCAm patients.
At this stage, the manuscript cannot be considered eligible for publication and requires major revisions.
We thank the reviewer for their thorough evaluation of our manuscript and for the constructive feedback provided. We acknowledge the reviewer’s concerns for substantial revisions been necessary before the manuscript can be considered for publication.
We have carefully reviewed each of the points raised and have undertaken significant revision to address the concerns in a comprehensive and thoughtful manner. Our responses are outlined on detail above along with a revised version of the manuscript that incorporates the suggested changes.
We believe that these revisions have substantially improved the quality, clarity, and scientific rigor of the manuscript. We are grateful for the reviewer’s insights, which have helped us strengthen our work, and we hope that the revised version will now meet the standards required for publication.

Round 2
Reviewer 2 Report
Comments and Suggestions for Authors
It is eligible for approval